# Spatiotemporal Patterns of Cholera Hospitalization in Vellore, India

**DOI:** 10.3390/ijerph16214257

**Published:** 2019-11-02

**Authors:** Aishwarya Venkat, Tania M. Alarcon Falconi, Melissa Cruz, Meghan A. Hartwick, Shalini Anandan, Naveen Kumar, Honorine Ward, Balaji Veeraraghavan, Elena N. Naumova

**Affiliations:** 1Friedman School of Nutrition Science and Policy, Tufts University, Boston, MA 02111, USA; Aishwarya.Venkat@tufts.edu (A.V.); Tania.Alarcon_Falconi@tufts.edu (T.M.A.F.); 2Sackler School of Graduate Biomedical Sciences, Tufts University, Boston, MA 02111, USA; melissa.e.cruz@gmail.com (M.C.); hward@tuftsmedicalcenter.org (H.W.); 3School of Marine Science and Ocean Engineering, University of New Hampshire, Durham, NH 03824, USA; mah2002@wildcats.unh.edu; 4Christian Medical College, Vellore, Tamil Nadu 632004, India; shalinianandan@cmcvellore.ac.in (S.A.); speed.naveen1@gmail.com (N.K.); vbalaji@cmcvellore.ac.in (B.V.); 5Tufts Medical Center, Boston, MA 02111, USA

**Keywords:** spatial statistics, disease clusters, cholera, hospitalization, India, electronic health records (EHR), mobile population

## Abstract

Systematically collected hospitalization records provide valuable insight into disease patterns and support comprehensive national infectious disease surveillance networks. Hospitalization records detailing patient’s place of residence (PoR) can be utilized to better understand a hospital’s case load and strengthen surveillance among mobile populations. This study examined geographic patterns of patients treated for cholera at a major hospital in south India. We abstracted 1401 laboratory-confirmed cases of cholera between 2000–2014 from logbooks and electronic health records (EHRs) maintained by the Christian Medical College (CMC) in Vellore, Tamil Nadu, India. We constructed spatial trend models and identified two distinct clusters of patient residence—one around Vellore (836 records (61.2%)) and one in Bengal (294 records (21.5%)). We further characterized differences in peak timing and disease trend among these clusters to identify differences in cholera exposure among local and visiting populations. We found that the two clusters differ by their patient profiles, with patients in the Bengal cluster being most likely older males traveling to Vellore. Both clusters show well-aligned seasonal peaks in mid-July, only one week apart, with similar downward trend and proportion of predominant O1 serotype. Large hospitals can thus harness EHRs for surveillance by utilizing patients’ PoRs to study disease patterns among resident and visitor populations.

## 1. Introduction

In regions lacking robust infectious disease surveillance mechanisms, health authorities can substantially benefit from utilizing detailed hospitalization records that typically include the patient’s demographic information, personal data, confirmed diagnosis, and history of illness [1,2,3]. With increasing computerization of medical records and diagnostic improvements, electronic hospitalization records (EHRs) can be effectively utilized for individualized treatment and healthcare management and as a tool for targeted disease surveillance [4,5,6,7]. In resource-poor areas, the use of EHRs for disease monitoring on local and regional scales could be of high value, especially if the patient profile, hospital capture geographic area, and sources of exposure are well understood. 

Three location-based pieces of information are relevant to source-tracking of a disease—the patient’s place of residence (PoR), the place of exposure (PoE), and the place of health care (PoH) (Figure 1). PoE is typically determined based on patient recall and epidemiological investigations and ideally could be recorded in medical history or EHR. PoR is likely to be reported by the patient during admission into a hospital, which is often the PoH. In some situations of mild infection, two or more of these locations may be the same. For example, if the individual consumes contaminated water collected from a well close to their home, suffers diarrhea, and self-medicates with over-the-counter antidiarrheal drugs, we might conclude that PoR, PoH, and PoE are the same. For severe cases that require medical assistance and hospitalization, both PoR and PoH are known, but there may be substantial uncertainty regarding PoE. Detailed PoR and PoE information are often collected in investigations following disease outbreaks and may not be part of a standard diagnostic questionnaire.

Individual EHRs may contain information about PoE, PoR, and PoH, as well as time-stamped information on laboratory-confirmed disease vector or infectious agents. These EHR-derived data can be combined to create a unique and complete picture of infectious disease exposure, manifestation, and treatment patterns. Detected patterns can be useful for characterizing a hospital capture area, for example, by defining the average distance between PoR and PoH, or for identifying hotspots of infections based on patient PoRs [5,8]. When PoE is not included in EHRs, PoH or PoR could be used as a proxy, but only after careful examination of spatial patterns of hospitalizations. 

The Christian Medical College (CMC) Hospital in the Indian city of Vellore in Tamil Nadu state, plays an important role in documenting the changing landscape of cholera on the Indian subcontinent and serves as a national reference laboratory. The Department of Microbiology at CMC was instrumental in detecting the first outbreak of cholera caused by the O139 serogroup. The outbreak started in Vellore in September 1992 and spread to Madras (now Chennai) by October 1992 [9]. This epidemic subsequently spread to Calcutta city in the Indian state of West Bengal, and the country of Bangladesh [9]. The new serogroup designation O139, synonym “Bengal,” became the most prevalent serogroup worldwide [10,11]. The Department of Microbiology at CMC has tracked the progress of *Vibrio cholerae (V. cholerae)* O139 since first detection, documenting the virtual absence of O1 *V. cholerae* during 1992–1993, its reappearance in late 1993, and the prevalence of both O1 and O139 *V. cholerae* serogroups in Vellore since then [12]. The O139 serotype has been transported around the world through trade and tourism and is now well-established in most South Asian countries [13,14]

Understanding the spatiotemporal patterns of cholera among patients at CMC is crucial for managing the hospital’s caseload given travel and migration patterns around Chennai and Vellore. Vellore is located near many major tourist destinations, and CMC Hospital is an important destination for medical tourism in the subcontinent. CMC is also located three hours from Chennai, the second most frequently visited destination for foreign tourists in India [15]. While foreign travel to Vellore peaks during January and February, domestic travel to both Chennai and Vellore peaks from October through December [16], coinciding with major Indian festivals such as Diwali, Dusshera, and Christmas. During these months, individuals are most likely to travel large distances through multiple transit modes to visit hometowns and relatives. Skilled employment is the primary motivation for out-migration from Chennai, especially to nearby centers such as Bengaluru [17]. There is also significant economic migration between Tamil Nadu and the nearby states of Karnataka, Kerala, and Andhra Pradesh [17]. Given this mobile population, hospital-based surveillance in this region provides valuable information on endemic diseases and novel pathogens [18]. 

Cholera is a highly variable disease driven by local environments as well as seasonal and community-level factors governing disease transmission. Toxigenic cholera has a median incubation period of 1.4 days, with 95% of cases developing symptoms within five days [19]. Due to the variety of drivers and quick onset, establishing exposure-disease associations for cholera at the individual level can be challenging [19]. In such situations, one can utilize point processes, or stochastic processes whose events or results are observed within a study area and treated as a realization of a random point process in two-dimensional space [20]. Point process methods are based on individual events in a study region and therefore offer a distinct advantage to standard epidemiological modeling methods that are based on data aggregated in space and time [20].

This study examined geographic patterns of cholera-related hospitalization records maintained by CMC Hospital in Vellore, Tamil Nadu State, India. These records were used to examine spatiotemporal patterns of cholera based on patient PoR during 2000–2014. We used laboratory confirmed clinical isolates of *V. cholerae* abstracted from CMC logbooks and electronic databases to generate the 15-year record of cholera at the hospital. We geocoded each patient’s self-reported town and region, and developed point process models to identify clusters of PoR. Identified clusters were then modeled to examine their temporal characteristics, including peak timing and disease trend. These models were studied in conjunction with temporal covariates including holidays and weekends to characterize temporal and demographic differences between the two clusters. 

## 2. Materials and Methods 

### 2.1. Data Abstraction and Geocoding

Over 1900 laboratory-confirmed records of cholera between 1992 and 2014 were abstracted from stored logbooks (1992–2004) and electronic databases (2004–2014) maintained by the Department of Microbiology at CMC [21]. Approvals for data use and analysis were obtained from the CMC Institutional Review Board and by the Tufts Institutional Review Board. A laboratory definitive isolation of *V. cholerae* O1, O139, or non-agglutinating serotypes was used as the case definition for cholera. Continuous information was available for 15 years (1 January 2000–31 December 2014; 5479 days) and included 1401 records. Each record included the following fields for each patient: Date of Hospitalization, Town, State/Region, Age, Sex, and Serotype, which we used to create a cluster population profile. We standardized the Date of Hospitalization to a year-month-day format and removed 31 records with missing Town or Region from the dataset. The Town and Region fields were also standardized to reflect the most recent spelling of each place and were geocoded using the ggmap [22] R package. This automated geocoding process provided a latitude and longitude corresponding to the centroid of the town extent reported by each cholera patient as their PoR. Cases that could not be geocoded in the first round were flagged and examined for spelling errors. The Town and Region information was corrected for obvious typos, and for ambiguous entries, the nearest Town and State name was found from Google Maps. After corrections, a second round of geocoding was implemented. Overall, 1366 (97.5%) of records were successfully geocoded. A flowchart of the two-step geocoding process is shown in Figure 2, and a map of the geocoded records is provided in Figure 3. R Version 3.5.1 [23] and RStudio Version 1.1.463 [24] software were used for all data processing and statistical analyses. 

### 2.2. Examination of Point Patterns

Based on geocoded patient PoRs, analysis was limited to patient PoRs in India and its direct neighbors (Bangladesh, Bhutan, Nepal, and Pakistan). The boundaries of these countries were downloaded using the maps R package [25] and converted to an observational window in the UTM 44N projection using maptools [26], rgeos [27], sp [20,28], and spatstat [29] R packages. Cholera intensity was defined as the number of cases per unit area using the complete geocoded dataset. A simple point pattern was created, and the kernel smoothed intensity function of the point pattern was mapped as an exploratory tool. The intensity was first modelled using a cross-validated bandwidth of the isotropic Gaussian kernel to minimize mean-square error and then by maximizing the log-likelihood cross-validation over all data points [30,31,32]. The default standard deviation of the isotropic Gaussian kernel, calculated based on the point pattern extent, was further adjusted by multiplying it by factors of 0.25, 0.50, 0.75, and 1.0. This preliminary investigation further defined the search for two clusters within the dataset. 

### 2.3. Point Process Modeling

A series of point process models with only two-dimensional spatial trend (x and y) were investigated to characterize the two clusters. Smoothed Pearson residuals and Akaike Information Criterion (AIC) were collected from each model. The three models with the lowest AIC were utilized to extract boundaries for the two identified spatial clusters—one around Vellore and one in eastern India including the Indian state of West Bengal and the country of Bangladesh (henceforth “Bengal”). Boundaries of the smoothed residual field equaling zero were extracted to define cluster boundaries from each model. Each case was then classified as belonging to either the Vellore or the Bengal cluster per extracted boundaries from each model. Cases consistently classified as belonging to one cluster across all three models were designated as stable clusters.

### 2.4. Temporal Modeling

A list of Indian national holidays [33] and holidays for the state of Tamil Nadu [34] were compiled for the study period. Holidays from Tamil Nadu represented culturally significant festivals, which may align with increased long-distance travel to Tamil Nadu. Each date in the dataset was coded for binary holiday and binary weekend (Saturday or Sunday). Here, 1565 days (28.6% of the study period) were classified as weekends, and holidays accounted for 888 days (16.2% of the study period). 

Cholera cases were then compiled into daily time series of counts for the two stable clusters. Negative binomial regressions were used to model cholera counts in each cluster. The models were built from a basic regression equation with only linear trend and expanded to include yearly seasonality, weekends, and holidays (Table 1). The estimates of *β*_2_ and *β*_3_ regression coefficients and their error values were used to calculate peak timing and its variance using the δ-method [35,36]. Overall model fit was assessed based on percentage of variability explained (VE) by each model, calculated as shown in Equation (1).
(1)$$\mathrm{VE}=\frac{\left(D_{0}-D_{r}\right)}{D_{0}}*100\%$$
where D_0_—null deviance and D_r_—residual deviance. VE were also compared to AIC values.

## 3. Results

### 3.1. Exploratory Analysis

The complete geocoded dataset contained 1366 laboratory-confirmed cases of cholera. Given the short incubation period of toxigenic cholera, the largest volume of PoR was expected in the vicinity of CMC Hospital, from Tamil Nadu and neighboring Andhra Pradesh and Kerala states. As shown in Figure 4 which presents case counts aggregated by state, this expectation holds true. However, a notable number of patients also had a PoR in the Indian state of West Bengal and in the country of Bangladesh, which are located hundreds of kilometers away from Tamil Nadu (Figure 5). 

### 3.2. Point Process Intensity

Cholera case intensity, or the number of cases per unit area, was examined to investigate whether the observed clusters are consistent. As seen in Figure 6, Vellore consistently appears to be an important cluster for all smoothing kernel values. The extent of this cluster increases with increasing kernel size. As the magnitude of adjusted bandwidths increases, a region of moderately high case intensity is also observed in the Eastern portion of the study extent, spanning Bangladesh and eastern Indian states (Figure 6c–f). The cross-validated bandwidth intensity maps (Figure 6a,b) display a very narrow radius of case intensity around Vellore, indicating that the large percentage of cases from Vellore may mask lower case intensities from other regions. 

### 3.3. Point Process Modeling

A set of 14 models representing several families of spatial trends were investigated to describe this dataset (Table 2). The three models with the lowest AIC identified two clusters—one around Vellore with 836–841 cases, and one in Bengal with 294–493 cases (Figure 7). Cases consistently classified as belonging to one cluster across all three models were designated as stable clusters. Thus, the stable Vellore cluster had 836 cases (61.2% of the geocoded dataset), the stable Bengal cluster had 294 cases (21.5% of the geocoded dataset), and 326 cases were not part of either cluster (17.3% of the geocoded dataset). The Vellore cluster has a large average radius of approximately 306 km, indicating that CMC Vellore hospital has a large local capture area around Tamil Nadu, Andhra Pradesh, and Kerala states. The stable Bengal cluster has a small average radius of approximately 194 km located much farther away from CMC Vellore (Figure 7).

### 3.4. Temporal Effects and Patient Profile

In Figure 8, time series of counts show the temporal dynamics of cholera cases for the two stable clusters, and the histograms reflect the distributional shape of the temporal process. The first bar of the histograms indicates a high proportion of days with no cholera cases (88% for Vellore and 95% for Bengal). The time series plots show that both clusters have prolonged periods with no cases and occasional spikes of higher counts. 

Approximately 14% and 15% of cholera cases were reported during a holiday and 22% and 25% during a weekend for the Vellore and Bengal clusters, respectively (Table 3). Patients in the Bengal cluster were on average older (34 compared to 23 years old in the Vellore cluster) and predominantly male (63% compared to 56% in the Vellore cluster). The proportion of patients with cholera serotype O1 and O139 were similar in both clusters (Table 3). The consecutive model building process shown in Table 4 demonstrates improvement in AIC and VE when adding the seasonality term for both clusters, and only for the Vellore cluster when adding the calendar effects (holidays and weekends). Results from the final negative binomial regression model (Model III) with trend, seasonality, and calendar effects are shown in Table 5. 

Both clusters show a steady declining trend with cholera cases decreasing on average by 0.5% every 30 days. The Vellore cluster shows approximately 31% less cases reported during weekends (95% CI: −42.96% to −15.52%), while the Bengal cluster does not show a significant weekend effect. Holidays have no clear effect on either cluster. The average peak timing of reported cases was in early to mid-July, one week apart (7 July for Vellore and 14 July for Bengal), with the Bengal cluster having a wider confidence interval for peak timing than the Vellore cluster (± 23 days compared to ± 16 days).

## 4. Discussion

Detailed EHRs with accurate case information and patient PoR allow public health practitioners and data analysts to better understand the profile of hospitalized population and monitor infectious disease patterns. Our study demonstrates that patients diagnosed with cholera and treated at CMC Hospital represent a wide range of residential locations (PoR). These PoRs can be classified into two distinct geographic clusters—Vellore (61% of cases) and Bengal (22%). We found that the two clusters differ by their patient profiles, with patients in the Bengal cluster being most likely older males who are traveling to Vellore [21]. Both clusters show well-aligned seasonal peaks in mid-July, only one week apart, and they also show the same proportion of predominant O1 serotype.

Travel and migration are complex phenomena which cannot fully explain the link between observed PoR in Bengal and PoH in Vellore. However, we hypothesize that PoR locations as far as the Bengal cluster may indicate established travel patterns. Per the 2011 Indian Census, 1% of the population in the Bengal cluster was born in Tamil Nadu, which may explain why patients from this region may have travelled to the Vellore vicinity [37]. Employment opportunities may also motivate travel between these regions [17]. Given a median incubation period of 1.4 days [19], we posit that long-distance travel (over 1000 km) with symptomatic cholera seems unlikely. This hypothesis is generally supported in the literature—in an assessment of cholera-related hospitalization for children under five in Bangladesh, rural distances to hospital are classified in groups of less than 3 km, 3–5 km, 5–7 km, and greater than 7 km [38]. Other studies report mean distance to hospital of 4.9 and 6.7 km [39], and a maximum distance of 16.8 km [40]. Given this range of reported distances, we conclude that patients would likely not travel more than a few kilometers to seek treatment for cholera. Therefore, distances of greater than 1000 km between Vellore (PoH) and Bengal (PoR) observed in our dataset lead us to suspect that place of exposure (PoE) for the visiting population is within the Vellore cluster boundary. However, sound conclusions on this topic require further inquiry and microbiological analysis that are outside the scope of the current study.

This presented investigation is subject to several challenges and limitations. One underlying challenge is that the detection of disease clusters is determined by the accuracy of the underlying data. In our study many of the records before 2004 were collected from paper logbooks and validated with data from one of two hospital databases. The address entries had to be checked for discrepancies such as misspelling or misspecification of town or region. These non-standard data entry methods led to several uncertainties regarding the abstracted fields. There were also discrepancies between the two databases in use at CMC. Some entries were duplicated with varying patient attribute details, and some others were not diagnosed with cholera per one database. Attempts were made to resolve discrepancies by using matched records if available and treating different entries of cholera (despite potential of being the same case stored in different databases) as different cases. However, the transitional nature of data migration to EHR databases makes it difficult to retrospectively analyze the accuracy of the records. 

Spatial uncertainty regarding patient PoR also directly affects observed cluster boundaries. Several geographic boundaries and place names have changed since the reported case date. For example, Calcutta was renamed Kolkata in 2001, Pondicherry was renamed Puducherry in 2006, and the Indian state of Uttaranchal was created in 2000 and renamed Uttarakhand in 2007. Several entries were corrected in the second round of geocoding (Figure 2) to address changing geographies. Despite these challenges, EHRs utilized in this dataset were quite robust. Over 97% of the dataset was geocodable with minor data cleaning, indicating that hospitalization datasets often have very high quality and fidelity. Since only the Town and Region for each patient was abstracted to protect patient privacy, there was also inherent spatial uncertainty in the exact location of each PoR. A few studies have utilized high spatial resolution information from hospitalization records for source-tracking [4,5,41]; however, we were unable to abstract this level of detail in our dataset. Utilizing the centroid of a town as the PoR was a useful but coarse assumption for this study. The accuracy of future studies can be improved by geocoding the complete patient address obtained with appropriate patient consent and IRB approval. Patient travel information contained in EHRs can also help characterize the PoE more precisely for a thorough investigation of disease transmission among and across communities [4]. 

Another limitation is that hospital-based surveillance only allows us to observe extreme cases that require intensive care. Such systems can only detect a narrow range of cases and do not capture less severe cases that did not require hospitalization, patients who cannot afford hospitalization, or patients who self-medicate. According to the PoR-PoE-PoH framework introduced in Figure 1, our analysis does not capture cases with missing PoH. Therefore, any epidemiological study utilizing hospitalization records cannot make conclusions about prevalence or incidence in the general population. Furthermore, hospitals have limited capacity, and we do not have information about upstream admission and testing decisions which led to each record observed in our dataset. However, we demonstrate that hospitalization records are extremely valuable for understanding mobile patient populations. Given the distribution of patient PoRs in this dataset, we conclude that CMC Hospital’s capture area for cholera includes a local population from Tamil Nadu, Andhra Pradesh, and Kerala states and a visiting population from the Bengal region. This information is particularly useful for hospital administrators making daily decisions regarding staffing, procurement of laboratory supplies, seasonal testing schedules, and other factors that affect the quality of patient care. Knowledge of patient demographics and disease-specific peak timing across mobile populations is also extremely important to effectively manage outbreaks that may quickly overwhelm even large regional hospitals.

Like most infectious diseases, cholera exposure and manifestation are determined by a complex set of factors. Our study offers a preliminary inquiry considering spatiotemporal properties of the cholera caseload in this region. This analysis would benefit from additional demographic, socioeconomic, and public health covariates in the model. Traditional variables used in epidemiological clustering analysis include meteorological characteristics temperature, humidity, precipitation; and individual patient characteristics age, sex, and water and sanitation access [42]. However, given the ambiguity in PoE, it is difficult to determine an appropriate spatial extent for environmental data extraction. While demographic characteristics can be abstracted at the district level, access to water and sanitation facilities is extremely spatially and demographically variant, and information on these facilities was not available at a scale relevant for this study. In the presence of better information, existing methods to develop local patient profiles from EHRs [21] can be supplemented with PoR and PoE information to improve surveillance efforts and help rapidly characterize susceptible populations in the event of an outbreak. Local authorities can also implement simpler surveillance solutions by similarly geocoding patient PoRs to characterize the burden of local and imported cases of any disease over time. 

Moving forward, we recommend permanent solutions to improve primary data collection across EHR fields through standardized data entry formats. For example, address entry can be streamlined by using dropdown menus for key fields instead of text boxes or manual entry systems. Hospital database interfaces could also display warning signs for incorrectly entered data and limit implausible values while entering patient age, sex, and town. These steps can establish a more streamlined data-to-results pipeline for active up-to-date surveillance results. With ongoing improvements in hospital data infrastructure, we expect rapid advances in the spatial and temporal resolution provided by EHRs for optimal local and regional infectious disease surveillance. Utilizing accurate PoR and PoE information in conjunction with the methods presented in this analysis can produce high-quality hospital-based predictive models of multiple locally significant diseases.

Improvements to the outlined methodology as well as primary data collection can help create an active surveillance network based on high-resolution spatial and temporal data. While this analysis only studies the hospitalization record of cholera from CMC Vellore, this methodology can be expanded to many more diseases and centers. EHRs can also be combined with health survey and morbidity data for a more complete picture of regional infectious disease burdens. As capacities for data storage and analysis increase exponentially, robust data standards can be used to develop a hospitalization record network, which allows epidemiologists to gather, analyze, and monitor large amounts of high-quality data at multiple scales. Such data must be curated in machine-readable format in secure and accessible data repositories to facilitate further research.

## 5. Conclusions

This study utilizes patients’ place of residence (PoR) and demographic information to identify two distinct clusters of patient origin based on EHRs at CMC Hospital in Vellore, India. Information regarding demographic characteristics and disease peak timings among resident and visitor populations is essential for effective care delivery and outbreak management. The PoR-PoH-PoE framework can be harnessed for disease source-tracking among mobile populations and for developing national hospital-based surveillance networks. Spatial and temporal models utilizing hospitalization records can thus provide valuable information for researchers, public health professionals, and decision makers about population profiles and patterns of infections.

## Figures and Tables

**Figure 1 ijerph-16-04257-f001:**
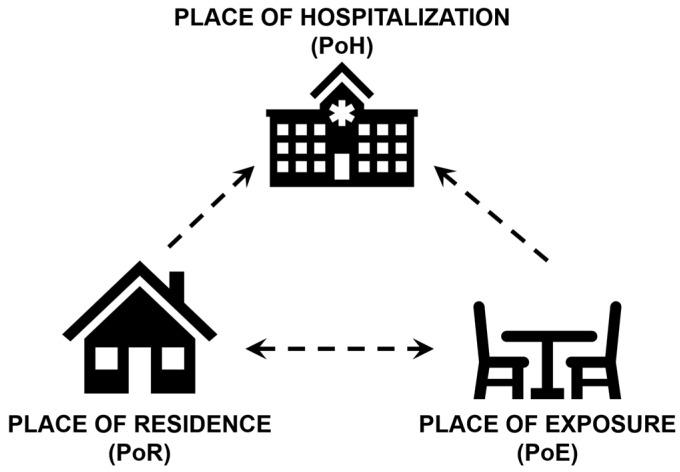
Location-based information relevant to disease source-tracking: Place of Residence (PoR), Place of Exposure (PoE), and Place of Hospitalization (PoH).

**Figure 2 ijerph-16-04257-f002:**
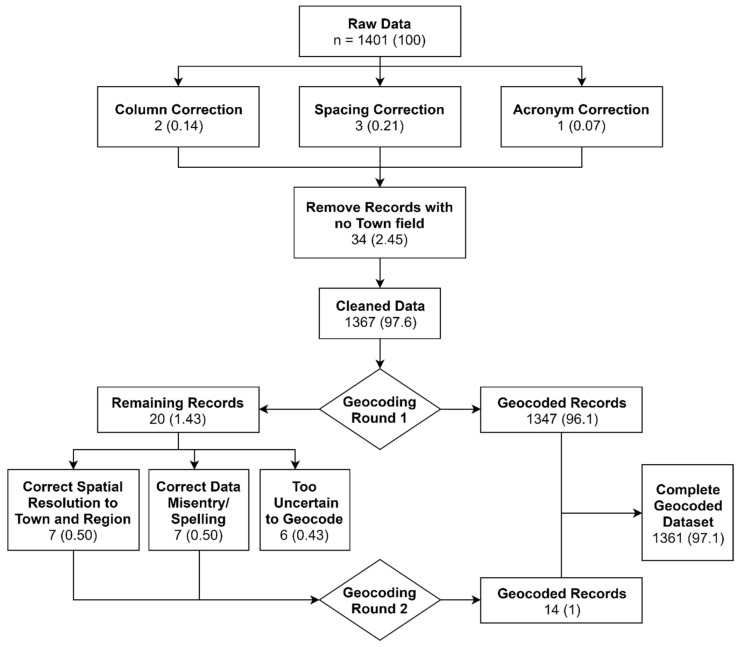
Flowchart of analytical process.

**Figure 3 ijerph-16-04257-f003:**
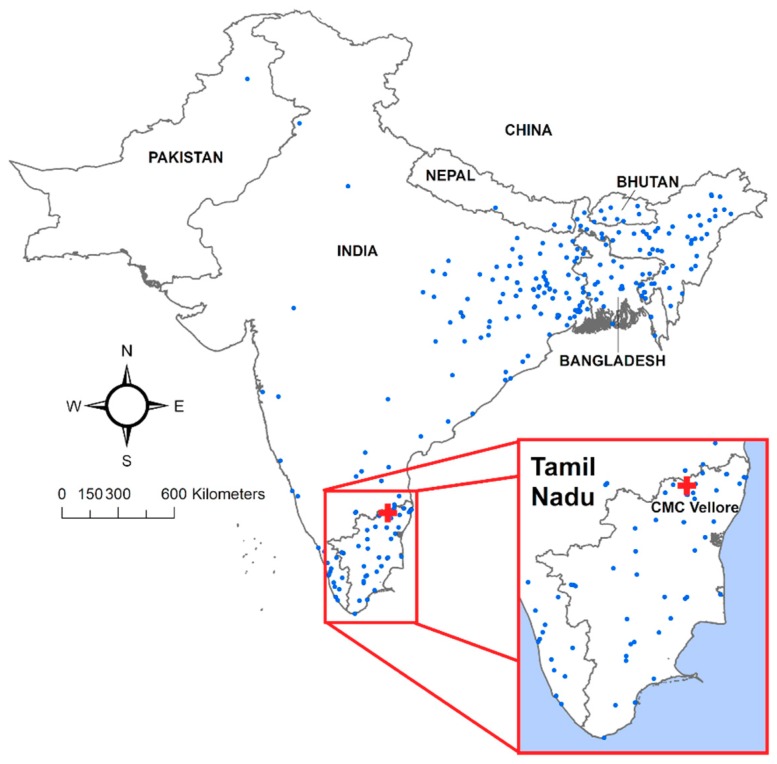
Map of cholera patients’ places of residence as observed at Christian Medical College (CMC) hospital in Vellore, Tamil Nadu state, India.

**Figure 4 ijerph-16-04257-f004:**
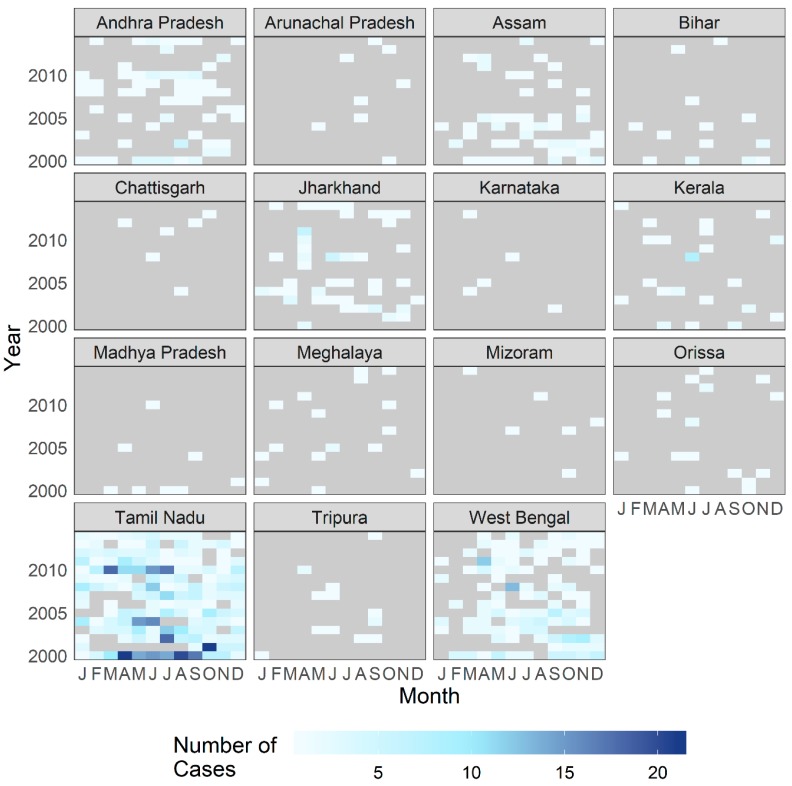
Number of patients treated for cholera at CMC Vellore by year and month, with reported place of residence by Indian state. The following states and territories are not shown due to low case counts (under five cases in complete dataset): Delhi, Goa, Mizoram, Nagaland, Punjab, and Sikkim.

**Figure 5 ijerph-16-04257-f005:**
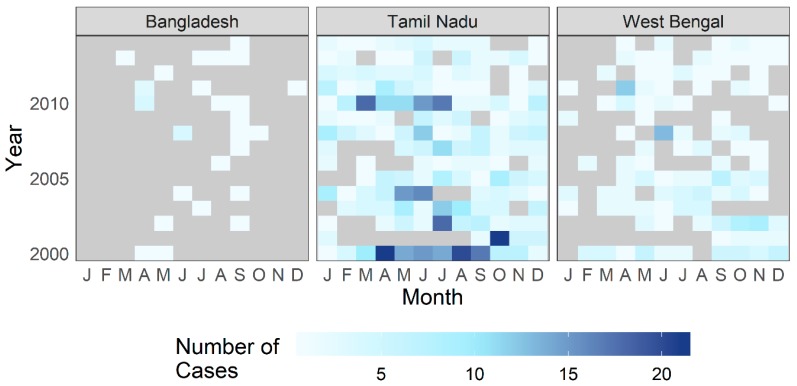
Number of patients treated for cholera at CMC Vellore by year and month from the country of Bangladesh and the Indian states of Tamil Nadu and West Bengal.

**Figure 6 ijerph-16-04257-f006:**
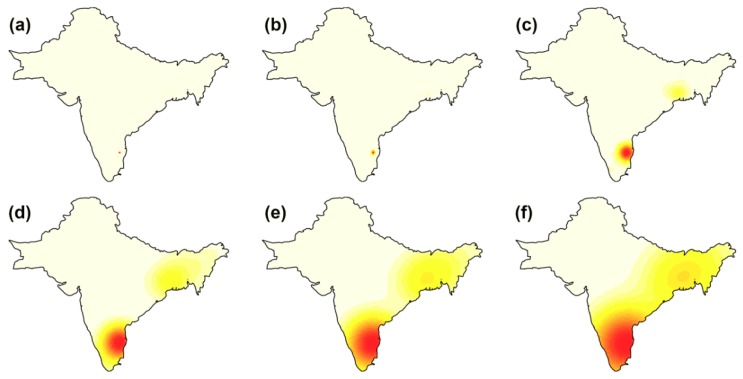
Isotropic Gaussian kernel smoothed intensity (cases per unit area) in CMC dataset. Each panel represents a bandwidth: (**a**) cross-validation; (**b**) likelihood cross-validation; and isotropic Gaussian kernel adjustment factors of: (**c**) 0.25, (**d**) 0.50, (**e**) 0.75, and (**f**) 1.

**Figure 7 ijerph-16-04257-f007:**
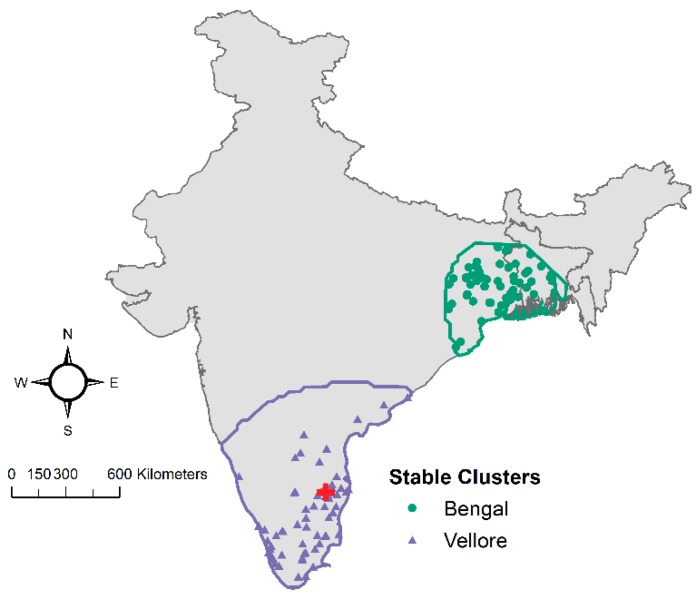
Boundaries and cases included in the Vellore (purple) and Bengal (green) stable clusters.

**Figure 8 ijerph-16-04257-f008:**
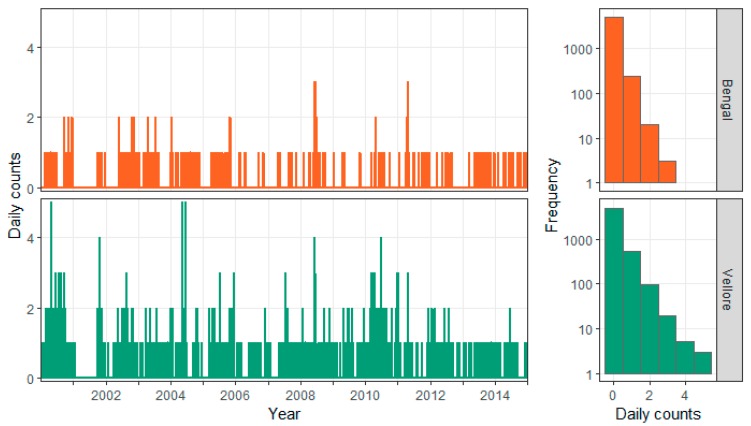
Daily time series and histograms for cholera cases in the stable Bengal (upper row) and Vellore (lower row) clusters.

**Table 1 ijerph-16-04257-t001:** Negative binomial models with trend, seasonality, weekends, and holidays.

Model	Model Formulation *
I: Trend	$ln\left(E\left[c\right]\right)=\beta_{0}+\beta_{1}t$
II: Trend + Seasonality	$ln\left(E\left[c\right]\right)=\ldots +\beta_{2}sin\left(2\pi \omega t\right)+\beta_{3}cos\left(2\pi \omega t\right)$
III: Trend + Seasonality + Weekends + Holidays	$ln\left(E\left[c\right]\right)=\ldots +\beta_{4}W+\beta_{5}H$

* Variables: *c* = cholera counts, *t* = days since the start of the study (e.g., *t* = 1, for 1 January 2000), ω = frequency calculated as 1/365.25, *W* = weekends, *H* = holidays.

**Table 2 ijerph-16-04257-t002:** Cases within Vellore and Bengal clusters based on spatial trend models.

Model No.	Trend	AIC *	Cases in Vellore Cluster	Cases in Bengal Cluster	Cases in Joint Cluster
15	sin(x) + cos(y)	58,132	836	493	-
14	cos(y) + x	58,351	841	294	-
8	cos(y)	58,475	836	470	-
11	sin(x) + y	60,160	800	500	-
13	cos(x) + y	60,244	798	500	-
3	y	60,536	800	500	-
12	sin(y) + x	61,123	-	-	1135
7	sin(y)	61,511	-	-	1327
9	sin(x + y)	62,028	-	-	1339
2	x	62,060	843	291	-
5	sin(x)	62,193	-	-	1339
6	cos(x)	62,214	-	-	1339
10	cos(x + y)	62,419	-	-	1339
1	1	62,558	-	-	1339

* AIC—Akaike Information Criterion.

**Table 3 ijerph-16-04257-t003:** Summary and effect of temporal covariates on case intensity for the stable clusters.

Variable	Measure	Vellore	Bengal
Total Cases	Count	836	294
Holiday	Count (%)	116 (13.88)	45 (15.31)
Weekend	Count (%)	181 (21.65)	72 (24.49)
Male	Count (%)	468 (56.05)	185 (62.93)
Age	Mean (Sd)	25.73 (23.14)	33.92 (18.81)
O1 serotype	Count (%)	574 (68.66)	180 (61.22)
O139 serotype	Count (%)	49 (5.86)	12 (4.08)

**Table 4 ijerph-16-04257-t004:** Summary of negative binomial model fit for the Vellore and Bengal stable clusters.

Model	Vellore		Bengal	
	AIC *	VE (%) **	AIC	VE (%)
I: Trend	4865	1.93	2308	1.02
II: Trend + Seasonality	4815	4.05	2288	2.81
III: Trend + Seasonality + Weekends + Holidays	2290	4.72	2290	2.96

* AIC—Akaike Information Criterion; ** VE (%)—percent variability explained.

**Table 5 ijerph-16-04257-t005:** Summary of trend, peak timing estimates, and weekend and holiday effects of cholera cases for the Vellore and Bengal clusters.

Variable	Vellore	Bengal
Trend as % change in disease counts per 30 days	−0.519 (−0.667, −0.371)	−0.455 (−0.690, −0.222)
Weekend effect as % difference	−31.27 (−42.96, −17.52)	−18.67 (−39.00, 7.31)
Holiday effect as % difference	−9.47 (−27.78, 12.73)	2.18 (−28.08, 42.33)
Peak timing in days	186.2 (170.5, 202.0)	194.6 (171.5, 217.8)
Peak timing	July 7 ± 16 days	July 14 ± 23 days

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
