# Peer review of "Spatiotemporal Patterns of Cholera Hospitalization in Vellore, India"

_ijerph, 2019, doi:10.3390/ijerph16214257_

Round 1

Reviewer 1 Report

Thank to authors for revising the manuscript. However, the resubmitted manuscript has not improved in content, conclusions are still speculative and too far fetched. Also, changes made in the manuscript, does not coincide with the claims provided in the rebuttal letter.  

Comments:

Title of the manuscript, does not reflect main idea of the manuscript; which emphasizes on the merits of electronic hospitalization records. Further, authors claim that they used EHRs and logbooks, but using terms EHR all along. It does not justify. Use of term, hospitalization record, would serve the purpose.  This reviewers does not agree with authors' claim that PoR-PoE-PoH information is used judiciously in source-tracking of the disease. Again, authors are speculating that "Given a median incubation period of 1.4 days [19], it is unlikely that a patient would travel more than a few kilometers to seek treatment for this illness.” There are not enough evidence provided to substantiate their claim. Authors could compare other diseases, with short incubation time where patient mobility is affected. Authors may like to avoid using too strong words and need to tone down their speculations, such as in the sentence given above. Authors should use either Bengal or West Bengal. Does Bengal cluster include country Bengladesh? It is not clear.  Manuscript need proofreading (e.g. line 126-127). For heatmaps, as stated in previous version, blue color is more obvious. A dark blue color could be interpreted as high or alarming. 

Author Response

Please note the changes in the text are highlighted in yellow.

Title of the manuscript, does not reflect main idea of the manuscript; which emphasizes on the merits of electronic hospitalization records.

We have modified the title to closer align it with the goals of our paper. Our goal as stated in the Abstract was to examine the “geographic patterns of patients treated for cholera at a major hospital in south India”. Therefore, we have replaced “clusters” with “patterns” and the new title reads as: Spatiotemporal Patterns of Cholera Hospitalization in Vellore, India.

Although we do discuss the value of electronic health records in capturing spatial information requisite for our analysis, this is clearly not the main idea of the manuscript. The core of our analysis is the examination of spatial and temporal patterns of patient origin at a hospital. We thus concur that the new title is sufficiently representative of our goals and analysis.  

Further, authors claim that they used EHRs and logbooks, but using terms EHR all along. It does not justify. Use of term, hospitalization record, would serve the purpose. 

We have modified this sentence in Section 2.1 to read: “Over 1900 laboratory-confirmed records of cholera between 1992 and 2014 were abstracted from stored logbooks (1992 – 2004) and electronic databases (2004 – 2014) maintained by the Department of Microbiology at CMC”. We have also included a reference to our previous paper on this dataset which reflects the same information.

Although the larger dataset contains cases from 1992 onwards, due to discontinuous data in 1992-1999, our analysis is limited to 2000-2014. This comprises our continuous time series of 15 years (January 1, 2000-December 31, 2014; 5479 days) and includes 1401 records. Thus, the first five years of our dataset were extracted from logbooks and the remaining ten years derive from electronic health records.

Electronic health records (EHRs) are the current global standard for storing patient information in clinical settings. In our dataset, both logbooks and EHRs are comparable (i.e. contain the same fields of interest). Since the majority and the most recent of our records derive from EHRs, we use the term “EHR” to describe all hospitalization records. We also discuss the challenge of data accuracy in the migration from logbooks to EHRs in the Discussion section.

This reviewers does not agree with authors' claim that PoR-PoE-PoH information is used judiciously in source-tracking of the disease. Again, authors are speculating that "Given a median incubation period of 1.4 days [19], it is unlikely that a patient would travel more than a few kilometers to seek treatment for this illness.” There are not enough evidence provided to substantiate their claim. Authors could compare other diseases, with short incubation time where patient mobility is affected. Authors may like to avoid using too strong words and need to tone down their speculations, such as in the sentence given above.

We accept the reviewer’s critique that this claim requires further references, and we have explored the literature to find hospital-based studies on cholera incidence that include a “distance to hospital” variable. Although we could not match the demographic and geographic context, we found several studies from Bangladesh that address this issue. These references have now been included in P2 of the Discussion section as follows:

“Given a median incubation period of 1.4 days [19], we posit that long distance travel (over 1000 km) with symptomatic cholera seems unlikely. This hypothesis is generally supported in literature—in an assessment of cholera-related hospitalization for children under five in Bangladesh, rural distances to hospital are classified in groups of less than 3 km, 3-5 km, 5-7 km, and greater than 7 km [38]. Other studies report mean distance to hospital of 4.9 and 6.7 km [39], and a maximum distance of 16.8 km [40]. Given this range of reported distances, we conclude that patients would likely not travel more than a few kilometers to seek treatment for cholera. Therefore, distances of greater than 1000 km between Vellore (PoH) and Bengal (PoR) observed in our dataset lead us to suspect that place of exposure (PoE) for the visiting population is within the Vellore cluster boundary. However, sound conclusions on this topic require further inquiry and microbiological analysis which are outside the scope of the current study”.

With this background, we believe our conceptualization and application of the PoR-PoE-PoH framework is sufficiently judicious.

Authors should use either Bengal or West Bengal. Does Bengal cluster include country Bengladesh? It is not clear. 

The relevant landmass surrounding the Bay of Bengal includes administrative boundaries of West Bengal state, located in India, as well as the country boundary of Bangladesh as shown in Figure 1. All references to this region are now clarified as the “Indian state of West Bengal and the country of Bangladesh”, or referred to explicitly as the Bengal cluster.  

Manuscript need proofreading (e.g. line 126-127).

This reference error was caused by editorial style formatting which affected cross-reference numbering. We have addressed these issues in the revised manuscript.

For heatmaps, as stated in previous version, blue color is more obvious. A dark blue color could be interpreted as high or alarming.  

We have updated Figures 4 and 5 in accordance with the reviewer’s color preference. The updated figure also removes states with sparse data (less than five cases in the complete time series). Figures 4 and 5 now provide a greater diversity of hues for a clear understanding of low vs. high case counts across states and months.

Reviewer 2 Report

The revised manuscript by Venkat et al. was a much-improved version of the original submission. All my previous queries and concerns were either answered or satisfactory explanation was provided. The authors have done a commendable job in this resubmission.

I have the following comments for minor revision.

Line 126: Check the error in the reference Line 191: …increasing kernel size. For higher-order values of σ, a region of relatively high case intensity is also…

The term “σ” has never been referenced before in the methodology. At least, something related should have been mentioned in section 2.2

Line 246-249: “The Vellore cluster has a large average radius of approximately 306 km, indicating a large local capture area around Tamil Nadu, Andhra Pradesh, and Kerala states for CMC Vellore hospital. The stable Bengal cluster has a small average radius of approximately 194 km located much farther away from CMC Vellore. “

These sentences should not be part of the discussion. These are new results that have not been previously presented in the results section. I will suggest you move it to the “results” and indicate the cluster areas with a circle (proportionately) on Figure 7.

Author Response

Comments and Suggestions for Authors: The revised manuscript by Venkat et al. was a much-improved version of the original submission. All my previous queries and concerns were either answered or satisfactory explanation was provided. The authors have done a commendable job in this resubmission.

Line 126: Check the error in the reference

The reference error was caused by editorial style formatting which affected cross-reference numbering. We have addressed these issues in the revised manuscript.

Line 191: “…increasing kernel size. For higher-order values of σ, a region of relatively high case intensity is also…” The term “σ” has never been referenced before in the methodology. At least, something related should have been mentioned in section 2.2

This lack of clarification for σ is duly noted. Since the actual standard deviation of the kernel is not the focus of our methodology, we have chosen to remove references to σ in the manuscript. We have corrected the point pattern intensity examination in Section 2.2 to read: “The default standard deviation of the isotropic Gaussian kernel, calculated based on the point pattern extent, was further adjusted by multiplying it by factors of 0.25, 0.50, 0.75, and 1.0. Resulting densities were mapped as shown in Figure 6”.

We have further removed the term “σ” in Line 191, since we are referring to the adjusted bandwidth and not the actual standard deviation in this sentence. This sentence now reads: “As the magnitude of adjusted bandwidths increases, a region of moderately high case intensity is also observed in the Eastern portion of the study extent, spanning Bangladesh and eastern Indian states (Figure 6 (c)-(f))”.

Line 246-249: “The Vellore cluster has a large average radius of approximately 306 km, indicating a large local capture area around Tamil Nadu, Andhra Pradesh, and Kerala states for CMC Vellore hospital. The stable Bengal cluster has a small average radius of approximately 194 km located much farther away from CMC Vellore. “

These sentences should not be part of the discussion. These are new results that have not been previously presented in the results section. I will suggest you move it to the “results” and indicate the cluster areas with a circle (proportionately) on Figure 7.

These sentences have been moved from Discussion to Results, and we have included a reference to Figure 7 as recommended. The boundaries shown in Figure 7 sufficiently capture this average radius.

Reviewer 3 Report

[Overall comment]

This study examined geographic patterns of cholera-related patients which were classified into two clusters of patient residence, Vellore and the West Bengal. Authors constructed spatial trend models during 2000-2014. I would like to suggest several comments to improve the main points and ask questions.

[Major comment]

Variability explained (VE) was defined in P5 line 168-9. But, it is not clear how to calculate VE. It can be described by using equations. Figure 4 is difficult to see and interpret the pattern due to mostly small number of patients, while Figure 5 shows the explicit pattern of the number of patients. I think it’s better to use discrete scale of the legend color for the number of cases to capture the patter even for the small number of patients.

[Minor comment]

P3 line 126-7: Citation of the reference had an error. P3 line 127: “R [22] and RStudio [23] software were used” It requires more detail information such as version of software authors used. P5 line 160: variables, “Holiday” and “Weekend” were expressed in italic. Model contents in Table 1 didn’t use italic texts for two variables. Please use the common styles. P5 line 170: “Model contents” in Table 1 and “Model” in Table 4 can use the same names to refer to each model. Table 4 was shown twice in P8 and P9. P8 Table 4: VE is abbreviation. Please write down the full terminology when first introduced. P9 Figure 8: I think bars represent the weekly counts, not daily counts. Does it show a total of 365 bars corresponding to a year? P9 Table 5: Table 3 and Table 4 showed the values rounded to two decimal places. Table 5 can be shown with the same number of decimal point. In reference 14 and 34, “pp.” should be removed.

Author Response

Variability explained (VE) was defined in P5 line 168-9. But, it is not clear how to calculate VE. It can be described by using equations.

Equation 1 was previously defined in-line. We have updated this to a standalone equation, with a description of each of the variables including VE.

Figure 4 is difficult to see and interpret the pattern due to mostly small number of patients, while Figure 5 shows the explicit pattern of the number of patients. I think it’s better to use discrete scale of the legend color for the number of cases to capture the patter even for the small number of patients.

Use of a discrete scale would be misleading for our dataset since there is a clear continuous difference in caseloads across state and time. We have updated the color scale for Figures 4 and 5 to provide a clear visual interpretation of light blue (less than approximately 10 cases) and dark blue (greater than approximately 10 cases). We have also excluded states and territories with low case counts from Figure 4 to present the most relevant information to the reader.

[Minor comment]

P3 line 126-7: Citation of the reference had an error. P3 line 127: “R [22] and RStudio [23] software were used” It requires more detail information such as version of software authors used.

We have added version information to both the reference and the sentence. This now reads: “R Version 3.5.1 [23] and RStudio Version 1.1.463 [24] software were used for all data processing and statistical analyses”

P5 line 160: variables, “Holiday” and “Weekend” were expressed in italic. Model contents in Table 1 didn’t use italic texts for two variables. Please use the common styles.

Formatting has been amended per reviewer’s recommendation.

P5 line 170: “Model contents” in Table 1 and “Model” in Table 4 can use the same names to refer to each model.

Table 1 has been modified to reflect the model descriptions in Table 4.

Table 4 was shown twice in P8 and P9.

This error was caused by editorial style applications and has been resolved.

P8 Table 4: VE is abbreviation. Please write down the full terminology when first introduced.

We have included Equation 1 and a complete explanation of variables in the preceding and following sentences.

P9 Figure 8: I think bars represent the weekly counts, not daily counts. Does it show a total of 365 bars corresponding to a year?

This graph represents a daily time series for the entire study period. There are 365 bars per year, but many days have zero cases.

P9 Table 5: Table 3 and Table 4 showed the values rounded to two decimal places. Table 5 can be shown with the same number of decimal point.

We have removed extra decimals when appropriate. Higher precision is required for Table 5, specifically for the effect size since values are less than zero.

In reference 14 and 34, “pp.” should be removed. 

Since both Reference 14 and 35 (34 refers to a website) refer to books, we have retained pagination information. However, we have removed the DOI information for the books in accordance with IJERPH Template References 2-3 and the MDPI Citations Style Guide.

Round 2

Reviewer 1 Report

Authors have addressed all the concerns raised sufficiently. I applaud their efforts and sincerity. 

This manuscript is a resubmission of an earlier submission. The following is a list of the peer review reports and author responses from that submission.

Round 1

Reviewer 1 Report

Manuscript ID: ijerph-581527

Thank you for inviting me to review this manuscript titled “Spatiotemporal Clusters of Cholera Hospitalization in Vellore, India.”

The idea of the manuscript is relevant to this journal and the applied methods are appropriate, however, I am confused about the study region and why it was generalized to the entire country. The authors said explicitly that their study represents only a single hospital in Vellore, Tamil Nadu, India. Therefore, their findings are explicit to this one area served by a single medical facility. From the methods, 1900 laboratory confirmed cholera cases only reflect the number of cases in Vellore city for 13 years, it is not representative of the entire subcontinent of India. The data only reflects the patient’s usual place of residence which should not be used as a proxy for incidence of cholera outside the study region or even the entire Tamil Nadu state. Perhaps the authors can extract cholera data from all Christian Medical College in India or restrict their study to Tamil Nadu state or carry out a temporal analysis only.

Other comments:

Line 19: This study examined geographic patterns of patients treated due to cholera at a major hospital in India.

Specify the exact study area in India. Is it Vellore, India?

Line 66-67: The authors said “This epidemic subsequently spread to Calcutta, Bangladesh, and the coastal area of the Bay of Bengal.” Make it clear to the readers that Calcutta and Bengal are in India and Bangladesh is another country. Also include a reference for this statement. Line 91-92: Confounding variables and low case counts further complicate building reliable statistical models for cholera [20]. Reference #20 is not related to cholera but on HIV. Section “2.2 Examination of Point Patterns.” Include a map of India indicating the study region showing neighbouring countries as well. Line 125-128: “Based on the patient geocoded locations, the study region for this analysis was India and its direct neighbors: Bangladesh, Bhutan, Nepal, and Pakistan.” This statement is confusing. Clearing state that the study area is Vellore, Tamil Nadu, India and refer to the map for neighbouring countries. Line 131-132: What kind of preliminary investigation was carried out to identify the two clusters? What are the two main methods used to calculate the kernel? Line 138-139: What are the specification of the series of models? Several Tables or Figures are not cited properly. For example, line 173, 181, 194, 196 and so on. Figure 2 is not appropriated, it is a bad choice for spatio-temporal visualization. You can either use a map or line graph. Where are the supplementary Figures S1 and S2? Figure 6 is better displayed with line graphs No new figure should be included or described in the discussion section, therefore removeFigure 7. There is no limitation of study.

Reviewer 2 Report

Manuscript submitted by Venkat et. al., entitled as “Spatiotemporal Clusters of Cholera Hospitalization in Vellore, India” analyzed hospitalization records of laboratory-confirmed cases of cholera from one hospital in India. Spatiotemporal clustering of cases helped authors to identify two different clusters of cases, onset of disease timing and based on those finding authors speculated that people are acquiring infection within the perimeters of hospital city. Presented study is an extension of authors' previous study (PLoS One 2017, 12, e0182642, doi:10.1371/journal.pone.0182642.), where authors had used exactly same dataset to analyze local patient profiles in order to characterize cholera in Vellore, India.

Study design is good and provide some insight towards disease surveillance in the area. But, most of the conclusions drawn are too speculative, naïve and without any significant analysis. E.g. Line 244-246, authors claim that since the season peak for disease from two clusters, identified in the study, separated only one week apart therefore travelers are likely to be infected in Vellore only. This conclusion is speculative, and without any conclusive evidence. It should be noted that O1 serotype had been prevalent in most of Indian peninsula, therefore its acquisition in a particular location need proper study. It also questions authors claim (line 296) to accurately define place of exposure using their statistical model.

Further, research study is retrospective. The model presented can only be used to analyze dataset in the past, however authors failed to provide any guideline or suggestion that how their model can be used to speculate disease prevalence in the future and how preventive measures and hospital preparation could be designed. Authors should have used their model to identify pattern of some other disease. It is not clear that their model is good for studying cholera only, which they identified to be in only two clusters; or is it equally applicable for other diseases which are endemic or pandemic in nature.

Authors could have supplemented dataset from other areas, e.g. Bengal; to support their findings.

Authors’ comments on streamlining data entry are appreciable and should be implemented.

Overall, presented study does not enhance our understanding of cholera epidemiology, or drivers governing disease spatiotemporal patterns. Neither it helps in developing a guideline to help policy makers in designing preventive measures.

Minor comments:

Several references are missing in the text. Line 300-301 need reference. Color of heatmaps in Fig. 2 and 3, should be changed. Yellow color (for fewer number of cases) is not obvious. Also, X-scale represent abbreviation for months should be stated in figure legend.